

# Transpressional tectonics during the Variscan-Alpine cycle transition: supporting a multi-rifting model, evidence from the European western Southern Alps

Emanuele Scaramuzzo[1], Franz A. Livio[1], Maria Giuditta Fellin[2,] Colin Maden[2]

[1]Department of Science and High Technology, University of Insubria, via Valleggio 11, 22100 Como, Italy
[2]Department of Earth Sciences, ETH Zürich, Sonneggstrasse 5, 8092 Zurich, Switzerland.

*Correspondence to*: M.G. Fellin (fellin@erdw.ethz.ch)

**Abstract.** We delve into the transition between the Paleozoic Variscan cycle and the Meso-Cenozoic Alpine supercontinent cycle, both of which have played a pivotal role in shaping the central European-Mediterranean plate's architecture. Our focus

is on the European western Southern Alps (Varese Area, N Italy), where we documented the tectonic events occurred during this transition. Two main scenarios have been proposed so far for this transition: i) a single, long-lasting, Permo-Triassic rifting event, culminating in the opening of the Alpine Tethys, or ii) multiple, distinct rifting events, preceding the onset of the Alpine cycle. By means of a tectono-stratigraphic and thermochronological approach, we recognized a first early Permian rifting stage associated with magmatic activity, followed during the early-middle Permian by transpressive tectonics and regional-scale

erosion that signal the end of the first cycle of crustal rifting. During the Middle Triassic, a second event initiated, which, we propose, marks the onset of the Alpine Tethys opening. This event could represent the stretching phase, which predates the well documented Upper Triassic crustal-thinning phase. Based on our findings, we propose that the Middle Triassic stretching phase represents the first stage of the Alpine Tethys rifting, thereby rejecting the hypothesis of a continuous Permo-Triassic long-lasting phase of extension.

## 1 Introduction

The Paleozoic Variscan cycle and the successive Mesozoic-Cenozoic Alpine supercontinent cycle (sensu Wilson, 1968; Wilson et al., 2019, and references therein) have shaped the architectural framework of the central European-Mediterranean plate (e.g., Stampfli and Kozur 2006; Ballèvre et al., 2020). Nonetheless, the transition between the two cycles is open to different interpretations, due to the large hiatus in the geological record spanning from the latest Paleozoic to the earliest

Mesozoic. As a remnant of the Variscan chain (Figure 1a), the European Southern Alps stand as an ideal study area to unravel the geodynamic processes governing the transition between the two cycles. In this region, three main events took place during this transition.

Firstly, the deposition of an upper Carboniferous siliciclastic cover on the Variscan metamorphic basement was followed by early Permian crustal thinning and cal-alkaline magmatism throughout the former peri-Variscan chain (e.g., Stampfli, 1996,



2000; Ziegler and Stampfli, 2001; Stampfli and Kozur, 2006; Cassinis et al., 2012; Cassinis et al., 2018; Ballèvre et al., 2020).
Then, a non-depositional, erosive phase cut through this succession all around the Mediterranean area during the early-middle
Permian (e.g., Cassinis et al., 2012; Gretter et al., 2013; Cassinis et al., 2018 and references therein). The resulting erosive
surface was later covered by discontinuous, upper-middle Permian to Lower Triassic continental to shallow water marine
sediments followed by discontinuous platforms and intra-platform basins successions (e.g., Bernoulli, 2007; Gaetani, 2010
and references therein). Finally, crustal thinning started in the Late Triassic (e.g., Bertotti et al., 1993).

Two main scenarios have been postulated so far to explain how the transition between the Variscan and the Alpine cycles
occurred: in the first scenario, the succession of events described above relate to a long-lasting, single-rifting event between
the demise of the Variscan cycle in the late Carboniferous and the onset of the Alpine one in the Late Triassic (e.g., Winterer
and Bosellini, 1981; Siletto et al., 1993; Zanoni and Spalla, 2018; Roda et al., 2019). In the alternative scenario, successive
and distinct rifting events (hereafter referred to as multi-rifting) intervened between the two cycles (e.g., Stampfli et al., 2002;
Ziegler and Stampfli, 2001; Stampfli and Kozur, 2006 and reference therein) and the break between the two cycles occurred
at the end of the transition, in the Triassic. According to the single rifting model, the early Permian tectonic phase represents
the onset of a single, long-lasting rifting event, under the same plate kinematic framework, culminating with the opening of
the Alpine Tethys (e.g., Winterer and Bosellini, 1981; Siletto et al., 1993; Zanoni and Spalla, 2018; Roda et al., 2019).
Conversely, according to the multi-rifting model, the Late Triassic opening of the Alpine Tethys was preceded by a series of
aborted rifting events, taking place under diverse plate kinematic directions (e.g., Ziegler and Stampfli, 2001; Stampfli and
Kozur, 2006, and references therein). Several models are concordant with the multi-rifting hypothesis, but each one proposes
different mechanisms to explain the succession of distinct rifting events from the early Permian to the Middle Triassic. For
instance, Muttoni et al. (2003; 2009) and Schaltegger and Brack (2007) concluded that multi-rifting reflects wrench tectonics
related to the reconfiguration of Pangea from an Irving-type (Pangea B) to a Wegener-type (Pangea A). Alternatively,
Malavieille et al. (1990) suggested that extension in the early Permian was caused by the orogenic collapse of the Variscan
chain. Finally, the early Permian extension could reflect back-arc thinning triggered by the subduction of the Paleotethys ocean
beneath the southeastern margin of the Pangea continent (Visonà, 1982, Lorenz and Nicholls, 1984; Stille and Buletti, 1987;
Di Battistini et al., 1988; Finger and Steyrer, 1990, 1991; Bonin et al., 1993; Doglioni, 1995; Stampfli and Kozur, 2006). A
combined tectono-stratigraphic and low-temperature thermochronological approach on a sector of the former Variscan chain
accreted within the western sector of the Southalpine chain provide novel constraints on the Variscan-Alpine cycles transition.
The pristine geological and thermochronological record of the study area shows that the Variscan-Alpine orogenic cycle
changeover is marked by a main aborted rifting phases, separated by a transpressive phase, thus supporting the multi-rifting
interpretation.



## 2 Geological background

### 2.1 Regional tectonic framework

The Southern Alps stand as a remnant of the Variscan chain (Figure 1a) that underwent rifting during the Mesozoic, ultimately transforming into the distant passive margin of the northern sector of the Africa Plate, e.g., Adria (e.g., Zingg et al., 1990; Bertotti et al., 1993; Handy et al., 1999; Bernoulli, 2007; Schalteger and Brack, 2007). This belt is composed of Adria derived units stacked along alpine south-verging thrusts, which accommodated reduced shortening in the west (Rosenberg and Kissling 2014), and which locally reactivated inherited structures as, for example, the Marzio Fault in the study area (Figure 1b, c; e.g., Scaramuzzo et al., 2022). The fragments of the Variscan chain exposed in the Southern Alps are made up of an assemblage of several tectono-metamorphic units (e.g., Diella et al., 1992; Boriani et al., 2003; Siletto et al., 1993; Boriani and Villa, 1997; Di Paola et al., 2001; Spalla et al., 2006). In the study area, these units consist of amphibolitic facies, para- and ortho-gneisses, and schists (Serie dei Laghi Unit; e.g., Boriani et al., 1990; Handy et al., 1999; Figure 3). They experienced peak metamorphism in the early Carboniferous (340-320 Ma) and cooled to moderate temperatures in the late Carboniferous (ca. 305 Ma), when they reached middle crustal depths, as documented by Rb-Sr age on white mica and biotite (Figure 1d; Schalteger and Brack, 2007, and references therein). The top of the Variscan basement is marked by a regional erosional surface, the Hercynian Unconformity (Handy et al., 1999). This is overlain locally by Middle Pennsylvanian (ca. 310 Ma) continental conglomerates (i.e., Manno Conglomerate, Figure 1d; Jongmans 1960; Casati, 1978; Cassinis et al., 2012). During the early Permian (i.e., between 285 and 275 Ma), mafic to acid plutonic rocks with mantle melt involvement intruded the peri-Variscan metamorphic rocks at lower to upper crustal depths (Figure 1d; Barth et al., 1994; Shaltegger and Brack, 2007; Berra et al., 2015; Cassinis et al., 2012; Karakas et al., 2019). This magmatic activity was accompanied by the deposition within caldera complexes of thick successions of volcanic products with calc-alkaline affinity together with clastic continental sediments (Collio, Orobic and Tregiovo basins and Athesian and Sesia Valley caldera; Figure 1d; Schaltegger and Brack 2007; Marocchi et al., 2008; Quick et al., 2009; Berra et al., 2015; Berra et al., 2016). In the study area, the lower Permian intrusive rocks consist of aplitic microgranites, granites, and quartz-bearing, NE-SW striking, micro-porphyric dikes, which altogether form a subvolcanic complex (i.e., Ganna Granitic Stock; Baggio and De Marco, 1960; Govi; 1960; Stille and Buletti, 1987; Bakos et al., 1990; Schaltegger and Brack 2007). The emplacement of the Ganna Granitic complex occurred at 281.34 ± 0.48 Ma (Schaltegger and Brack, 2007), near the surface, at pressures most likely in the range of 0.5 – 0.75 kbar and certainly less than 2 kbar (Bakos et al., 1990) implying a maximum thickness of the units overlying the intrusive complex of 2-5 km. Tuffs, ignimbrites and andesitic to dacitic lavas with thin intercalations of clastic rocks of lower Permian age are extensively exposed in the study area (Baggio and De Marco, 1960; Govi; 1960; Stille and Buletti, 1987; Bakos et al., 1990; Schaltegger and Brack 2007).



**Figure 1: Regional geological setting and stratigraphy: a) distribution of the Palaeozoic peri-Variscan terrain in the Mediterranean area (modified after von Raumer et al., 2002); b) simplified geological map of the Southern Alps modified after Bigi et al., 1990; c) crustal-scale cross-section along the western sector of the Southern Alps (redrawn after Scaramuzzo et al., 2022); d) chronostratigraphic scheme of Carboniferous-Jurassic succession of the Southern Alps with the sequence subdivision adopted in this paper for the study area (redrawn after Schalteger and Brack, 2007;Berra et al., 2009; Cassinis et al., 2012; Gretter et al., 2013; Beltrando et al., 2015; Cassinis et al., 2018).**



Across the Southern Alps, a middle Permian erosive surface (the "middle Permian Unconformity", hereafter; Figure 1d) lies over the top of the exhumed composite metamorphic basement and the lower Permian succession, marking a depositional hiatus of ca. 15 to 20 million years (Figure 1d; e.g., Bernoulli, 2007; Schaltegger and Brack 2007; Gaetani, 2010; Cassinis et al., 2012; Gretter et al., 2013; Cassinis et al., 2018). At middle crustal levels, pervasive extensional deformation is recorded
during the middle-late Permian by the metamorphic basement north of the Como Lake (Real et al., 2023). The middle Permian Unconformity is overlain by a diachronous sedimentary wedge consisting of continental siliciclastic deposits and minor coastal marine sediments, which spans from the uppermost middle Permian to the Lower-Middle Triassic (Figure 1d; i.e., Verrucano Fm. and Servino Fm.; e.g., Bernoulli, 2007; Gaetani, 2010; Cassinis et al., 2012; Cassinis et al., 2018). During the Anisian stage, mixed carbonate/fine siliciclastic deposits were deposited over an extensive area (e.g., Gaetani et al., 1988; Brack and
Rieber, 1993; Bernoulli, 2007; Gaetani, 2010). In the study area, the Anisian siliciclastic deposits consist of micro-conglomerates with centimetric rounded quartz clasts dispersed in a red matrix, and of thin-bedded siltstones and fine sandstones with plane-parallel or cross laminations (i.e., Bellano Fm; Figure 1d; Stockar et al., 2013). At the middle-upper Anisian boundary, siliciclastic deposition ended throughout the Southern Alps and substantial carbonate platform grew in proximal areas and alternated with intra-platform basins, which hosted anoxic sediments at once with volcanic activity (Figure
1d; e.g., Gaetani et al., 1988; Brack and Rieber, 1993; Bernoulli, 2007; Gaetani, 2010; Stork et al., 2019; De Min et al., 2020). During this stage, the local formations consist of dolomitic limestones that were deposited in reef and shallow-sea environments (Dolomia di San Salvatore) and that are laterally heteropic with intra-platform limestones and anoxic shales (Meride Limestones and Besano Shales; Figures 1d; Bernoulli 1964; Zorn, 1971; Furrer, 1995; Bernoulli, 2007; Stockar, 2010; Stockar et al., 2013; Renesto and Stockar, 2018). The end of the Carnian is marked across the Southern Alps by the deposition
of terrigenous, evaporitic, and lesser carbonate sediments associated with sea-level fall together with the input of siliciclastic and volcanic detritus (Figure 1d; i.e., Pizzella Marls, San Giovanni Bianco Fm., Raibl Fm.; e.g., Bernoulli et al, 2007; Gaetani et al., 2010). The late Anisian to early Carnian development and emersion of carbonate platform was simultaneous with a thermal event recorded by the basement rocks north of the Como Lake, which underwent static recrystallization (Real et al., 2023). During the latest Carnian, the subsidence rate gradually decreased, and the volcanic activity ceased (e.g., Bernoulli,
2007; Gaetani, 2010). Subsidence resumed in the Norian and increased rapidly as crustal thinning progressed (e.g., Bertotti et al., 1993; Gaetani et al., 2010).

## 2.2 Thermochronological record

The thermochronologic record of the units accreted within the Southalpine chain (Figure 2) spans from the Carboniferous Period to the Miocene Epoch. Most commonly, the lowest temperature (low-T) thermochronometric data, including fission-
track and (U-Th)/He ages on apatite and zircon, record the thermal overprint related to the Mesozoic and Cenozoic events of the Alpine cycle (Giger, 1991; Martin et al., 1998; Bertotti et al., 1999; Zattin et al., 2006; Siegesmund et al., 2008; Zanchetta et al., 2011; 2015; Berger et al.; 2012; Pomella et al., 2012; Reverman et al., 2012; Wolff et al., 2012; Bertrando et al., 2015; Heberer et al, 2016). Higher temperature (high-T) thermochronometric systems, like Rb/Sr, K/Ar, and Ar/Ar, record also older



events that locally date back to the end of the Variscan cycle during the Carboniferous (Figure 2a, b; McDowell and Schmid,
1968; McDowell, 1970, Bertotti et al., 1999; Siegesmund et al., 2008; Zanchetta et al., 2011; Wolff et al, 2012). In the central
sector of the Southern Alps area, near lake Como, Jurassic to Early Cretaceous zircon fission-track ages (Figure 2c) indicate
continued cooling from rifting until the onset of Alpine convergence, whereas Late Triassic to Jurassic high-T
thermochronometric ages (Rb/Sr and Ar/Ar) document altered thermal conditions before and at the onset of rifting (Bertotti et
al., 1999). In the western sector of the Southern Alps, west of Lake Maggiore, apatite fission-track ages are all Eocene and
younger, whereas fission-track and (U-Th)/He ages on zircon are Triassic to Miocene, and high-T thermochronometric ages
(muscovite and biotite K/Ar; Figure 2a, b) are Carboniferous to Jurassic (McDowell and Schmid, 1968; McDowell, 1970;
Hurford, 1986; Siegesmund et al., 2008; Wolff et al 2012). In the westernmost sector of the Southern Alps, bedrock and detrital
zircon (U-Th)/He ages (Figure 2d) record a protracted thermal history that starts in the Late Triassic in relation to crustal
thinning and continues until the Cretaceous resulting in wide and complex age distributions in both bedrock and detrital
samples (Beltrando et al., 2015).

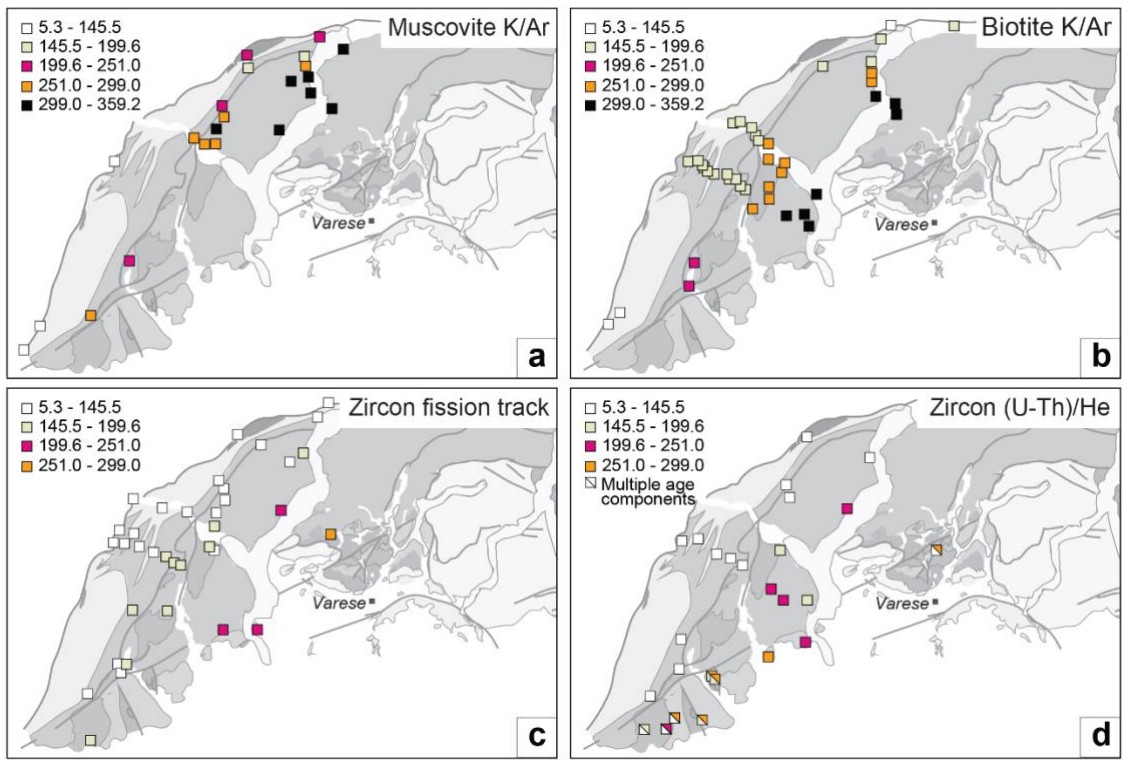

**Figure 2: a) Muscovite K/Ar ages compiled from McDowell, 1970; Hunziker, 1974; Zwingmann, 2004; Wolff et al., 2012; b) Biotite
K/Ar ages compiled from Jager and Faul, 1960; Carraro and Ferrara, 1968; McDowell, 1968; McDowell, 1970; Hunziker, 1974;
Hurford, 1986; Siegesmind et al., 2008; Wolff et al., 2012; c) Zircon fission track ages compiled from Hurford, 1986; Bertotti et al.,**
**1999; Siegesmund et al., 2008; Wolf et al., 2012; d) Zircon (U-Th)/He ages compiled from Wolff et al., 2012; Beltrando et al., 2015.
Sample with multiple age components are from detrital rocks and partially reset basement rocks (Beltrando et al., 2015).**



Ultimately, among the previous low-T thermochronometric data of the Southern Alps, the few Permian ages so far recorded are from two groups of samples consisting mostly of volcanic and detrital rocks and including one basement sample. One group of samples is located south-west of Lake Maggiore, where Permian volcanics and Jurassic sandstones give zircon (U-Th)/He ages that scatter over a very long temporal interval from the latest Carboniferous to the Late Cretaceous (Beltrando et al., 2015), indicating a low-temperature, protracted thermal history that causes variable age-resetting conditions. The second group of samples is in the Varese area and includes a 256 Ma-old zircon fission-track age in a schist (Giger, 1991) and a zircon (U-Th)/He age cluster between 300 Ma and 250 Ma combined with a Late Cretaceous age cluster in a Permian volcanoclastic rock (Beltrando et al., 2015). Thus, also this second group of samples has a complex thermochronologic record straddling age-resetting conditions.

## 3 Methods

### 3.1 Tectono-stratigraphy and thickness analysis

Starting from a geological map at 1:10,000 to 1:25,000 scale (after Scaramuzzo et al., 2022, modified), we focused on the tectonostratigraphic evolution from the early Permian to the Middle Triassic. We adopted the tectono-stratigraphic successions of the Southern Alps as proposed by previous authors (e.g., Bernoulli, 1964; Bertotti et al., 1993; Gaetani et al., 1998; Berra et al., 2009; Cassini et al., 2012). From the thickness-analysis, we derived an isopach 2D map contouring at the top of the Middle Triassic. This analysis is based on a 3D geological model, which is built with the MOVE software (courtesy of Petroleum Experts Ltd) and the restoration of the main Alpine structures. The geometry and orientation of geological structures were extrapolated in three-dimensions starting from a series of geological cross-sections and were integrated with constraints from both the surface trace of planar features and direct field measurements. We digitized the tops of the stratigraphic sequences on a series of geological cross-sections and then we interpolated surfaces across sections by means of a spline interpolation tool: this approach has been chosen to minimize surface curvature far from fault zones and to create a smoothed and simplified 3D geological model. Faults and horizons were created as 3D meshes. Finally, the 3D model was used to analyse the progressive deformation of the study area, through a sequential kinematic restoration. We divided the model into structural blocks, bounded by major faults. Each block was progressively restored by means of a 3D flexural slip unfolding algorithm to key horizons that were assumed as horizontal and arbitrarily fixed at zero meters above sea level. We adopted a 3D simple shear unfaulting method to restore the displacement of faults (Withjack et al., 1995). The restored geometry of faults was then compared to structural data collected on fault surfaces during structural analysis.

### 3.2 Kinematic analysis

Kinematic data were collected along the early Permian-Middle Triassic fault systems to determine the orientation of the best-fit moment tensor solution, enabling to calculate the paleo-stress orientation for each tectonic phase. Each group of data was inverted following a kinematic approach and assuming an overall correspondence between strain and stress axes. We calculated



moment tensor summations of P and T axes (i.e., E1≧E2≧E3; extension positive), as implemented in FaultKin 8

(http://www.geo.cornell.edu/geology/faculty/RWA/RWA.html). For a complete discussion of the assumptions and limitations

of these methods see Marrett and Allmendinger (1990). We inverted the fault slip data from their present-day orientation and,

in order to take into account the later Alpine tectonics, restored the obtained fault plane solution according to the bedding plane

(S0), extrapolated from the nearest sector, along strike.

### 3.3 Thermochronology

**3.3.1 Zircon (U-Th)/He dating**

We collected samples from gneiss and micaschists of the Variscan basement in the hanging walls and footwalls of major faults.

Zircon (U–Th)/He (ZHe) analysis was conducted at ETH Zurich. Zircon grains were first extracted from crushed samples

using conventional methods of heavy liquid and magnetic separation. Euhedral zircons with widths of >60 μm were then

selected from each sample under a polarized stereo-microscope. Each grain was photographed to measure its dimensions for

the calculation of alpha-ejection correction factor (FTK) following Ketcham et al. (2011). Afterwards, each zircon was packed

in Niobium foil, and loaded into an ultra-high vacuum sample chamber. $^4$He amounts were determined by outgassing with a

diode laser at 1090 ℃ for 45 minutes and measuring the released gas on a magnetic sector-field mass spectrometer equipped

with a Baur-Singer ion source in static vacuum. A second extraction was conducted for each grain at 1110 ℃ for 22 minutes.

A third extraction was performed at 1130 ℃ for 22 minutes for the grains, which still had high fractions of $^4$He released from

the second extraction (>1 %). Outgassed zircons were then transferred into Teflon vials, spiked with $^{233}$U and $^{230}$Th mixed

solution, and dissolved first at 225 ℃ for 72 hours in a concentrated mixture of HF and HNO$_3$ and then in concentrated HCl

at 200 ℃ for 24 hours to ensure dissolution of refractory fluoride salts. U and Th concentrations of each final solution were

measured on an inductively coupled plasma mass spectrometer (ElementXR). Six zircons from the Fish Canyon Tuff were

processed together with our samples and yielded a mean FTK age of 27.2 ± 1.5 Ma (± standard deviation (1s); supplementary

material Table S1), which are consistent with the recommended eruption age of 28.0–28.2 Ma (Boehnke and Harrison, 2014).

**3.3.2 Thermochronologic modelling**

To evaluate the thermal effect related to the granitic intrusion occurring in the area, we calculated the heat transfer from the

cooling granitic stock to the surrounding crust by means of a simple 1D transient diffusion equation. We assumed that the

magmatic body emplacement occurred instantaneously and that the size of the stock was relatively small with respect to the

thickness of the intruded crust (Ehlers et al., 2003). With these conditions the temperature (T) of any point at a given distance

from the intrusion (z) with time (t), is provided by Carslaw and Jaeger (1959):

$$T(z,t) = T_b + \frac{T_i - T_b}{2}\left[\mathrm{erf}\left(\frac{L/2 - z}{2\sqrt{\alpha t}}\right) + \mathrm{erf}\left(\frac{L/2 + z}{2\sqrt{\alpha t}}\right)\right]$$





where $T_i$ is the temperature of the intrusion, $T_b$ the one of the crust, L is the intrusive body thickness, and $a$ is the thermal conductivity coefficient. Erf is the error function (Abramowitz and Stegun, 1970).


## 4 Results

### 4.1 Tectono-Stratigraphy

The succession outcropping in the study area is comprised between the Carboniferous and the Cretaceous (Figure 3). We focused on the early Permian-Middle Triassic succession as it records the transition between the Variscan and the Alpine

cycles. Tectono-stratigraphic results are summarized in the following sections and illustrated in the geological map in Figure 3 and in the sections and tectono-stratigraphic scheme of Figure 4.





**Figure 3: Geological map of the Study area (modified after Scaramuzzo et al., 2022). Fault codes: MF, Marzio Fault; MBF, Martica - Boarezzo Fault; VGFr, Valganna Fault ramp segment; VGFf, Valganna Fault flat segment. The Mt. Nudo basin is Jurassic in age (Kalin and Trumpy, 1977).**

### 4.1.1 The lower Permian succession

The lower Permian succession is composed of: i) the Ganna Granitic Stock, intruded at shallow levels within the Variscan basement, with its intrusive contacts exposed both at the top and at the base of the stock (Figures 3 and 4); and ii) the effusive and volcano-clastic units, lying on top of the basement, above the Hercynian unconformity.

The lower Permian succession records the activity of the Marzio Fault. This is a near vertical structure that divides the study area into northern and southern blocks along a NE-SW direction (Figure 3). Our new observations along this fault indicate an





abrupt increase in the thickness of the lower Permian volcanic/volcanoclastic rocks with a change in thickness from ca. 200 m in the northern fault block to up to ca. 1500 m in the southern one (section A in Figure 4a, Figure 4b). This fault bounds sharply to the north also the lower Permian intrusive stock and puts it in contact with pre- to lower Permian, metamorphic and volcanic

rocks (Figure 3; Figure 4a, b). To the south, the lower Permian intrusive and extrusive rocks are also displaced by a reverse fault that we describe here for the first time, namely the Martica-Boarezzo Fault. This is a SW-NE striking, high-angle, SE-dipping fault. The Martica-Boarezzo Fault and the Marzio Fault bound a narrow push-up ridge (Mondonico push-up; Figure 3). This structure displaces the Ganna Granitic Stock onto the lower Permian extrusive units (sections A and C, in Figure 4a) and abuts onto the Marzio Fault to the north-east (Figure 3).

The middle Permian Unconformity seals the Martica–Boarezzo Fault (Figure 3 and Figure 4b). No data constraining the activity of the Mondonico push-up is available due to erosion at the top of the push-up; nonetheless, it is kinematically compatible with the Martica–Boarezzo Fault and shows the same crosscut relationships. The age of the activity of the Martica–Boarezzo Fault and of the Mondonico push-up can thus be constrained between the latest early Permian and the middle Permian (i.e., before the development at this site of the middle Permian regional unconformity).

**4.1.2 The Early-Middle Triassic succession**

On top of the middle Permian unconformity, thin, siliciclastic, Lower Triassic layers show no significant cross-cut relationship with the tectonic structures in the study area and are overlain by a thick carbonate succession that is Anisian–Ladinian in age. This succession consists of platform and intra-platform facies with a highly variable distribution. In particular, the intraplatform anoxic facies (i.e., Besano Shales) pinch out to the west. At the top, this succession is closed by an almost continuous interval

with constant thickness, composed by thin bedded evaporitic dolostones of Carnian age (Pizzella Fm. in Figure 1).
The Middle Triassic succession is gently folded or involved in Alpine thrusts north of the Marzio Fault, whereas, to the south, it is displaced by the Valganna Fault (Figure 3) with an apparent left-lateral separation. This structure is a normal fault presently showcasing an arcuate trace in map view due to Alpine tilting and erosion. Presently, the Valganna Fault is split into two segments with different orientation: a southern one, sub-vertical and striking N-S (VGFr in Figure 3 and sections B and C in

Figure 4a), and a northern one, dipping ca. 60° with an ENE-WSW strike (VGFf in Figure 3, and section D in Figure 4a). If back-tilted to their original attitude (S0 N170/40, measured ca. 1 km south of the VGFf), the two segments represent the shallow ramp (VGFr) and a slightly deeper and less dipping sector (VGFf) of the Valganna Fault, respectively. The apparent left-lateral separation is due to normal displacement. In fact, VGFr displaces the Anisian–Ladinian succession that shows a thickness increase from 500 m, to the east, up to 800 m, to the west (sections B and C in Figure 4a and c). VGFf presently dips

at a high angle to the SSE (sections C and D in Figure 4a) but its restored dip is close to 30°. It displaces the Ganna Granitic Stock, in the hanging wall, against the basement, in the footwall (Figure 3, section D in Figure 4a). At the junction between VGFf and VGFr, the Valganna Fault negatively inverts the inherited Martica–Boarezzo Fault. To the south, the Valganna Fault is clearly sealed by the top of the Anisian-Ladinian succession (Figure 3).



The Middle Triassic isopach map (Figure 4c) shows a variable basin topography in the study area. Differential subsidence, up
to 600 m, occurs across the Marzio Fault between the northern block (block B in) and the more subsiding southern block
(Block A in Figure 4c). The southern block features two narrows NW-SE elongated basins indicated as A1 and A2 in Figure
4c. The thickness analysis indicates that A1 and A2 are separated by a horst, are bounded by faults on the west and on the east
and accommodated up to 800 m thick, Anisian-Ladianian sediments. The thickness analysis also suggests that the Valganna
Fault bounds the eastern basin (A2) on the west and on the north, and the Marzio Fault bounds the western basin (A1) to the
north.



**Figure 4: a) Geological cross sections across the Study area; traces of the cross-sections in Figure 4. Codes: MBF, Martica-Boarezzo Fault; MNF; Monte Nudo Fault; MF, Marzio Fault; VGF, Valganna Fault; b) Tectono-stratigraphic block diagram of area (restored at Carnian); c) isopaches map of the Middle Triassic sequence (Anisian–Ladinian), after restoration, outcropping faults are represented with red lines.**





## 4.2 Fault kinematics and field evidence

In the field, the faults described in section 4.1 display the same crosscut relationship derived from mapping in several locations, including still-preserved kinematic indicators.

We measured kinematic data i) along the Martica-Boarezzo Fault where it cuts through the Ganna Granitic Stocks and the overlying Permian calc-alkaline succession (structural stations 1 and 2 in Figure 3), ii) along the Valganna Fault across the Ladinian carbonate platform rocks, and iii) along the contact between the basement and the Ganna Granitic Stock (structural stations 3 and 4 in Figure 3).

Along the Martica-Boarezzo Fault, fault steps and slickenlines indicate a reverse dip-slip sense of motion (Figure 5a, b) and

sets of P- and R-Riedel shears with related slickenlines that indicate right-lateral transpressive movements (Figure 5c, d). These kinematic indicators are associated with syn-kinematic quartz veins and porphyric dikes (Figure 5c, e). At places, reverse faults show evidence of a negative inversion, with overprinting steps and slickenlines, or a clear crosscut relationship with younger normal faults, displacing the reverse ones (Figure 5c).

The inversion of fault-slip data (Figure 7b, c) indicates a strike slip fault plane solution for the whole dataset, with a Shmax

(i.e., strain axis E3) oriented ca. NW-SE (Figure 7b, c). The inversion (Figure 7b, c) is consistent with the transpressive fault architecture as observed from mapping and field data.



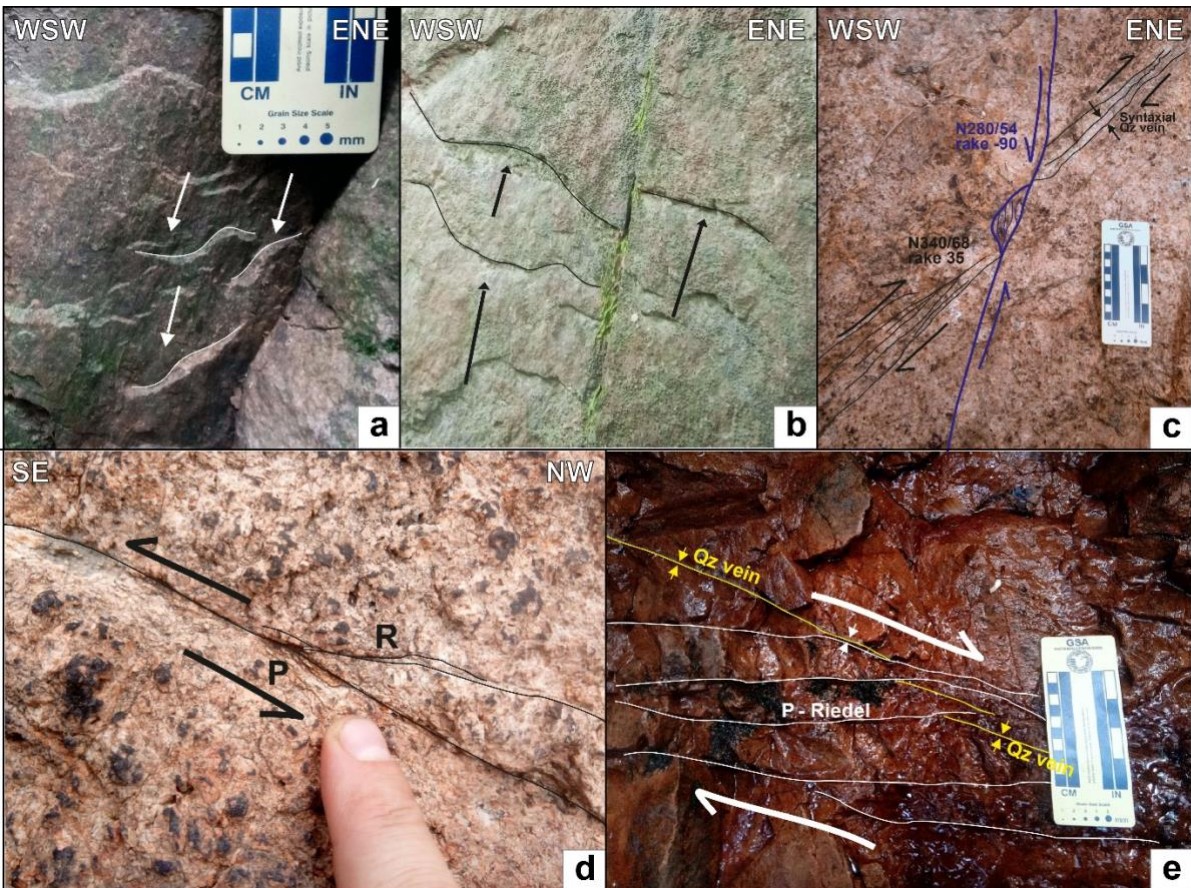

**Figure 5: Representative kinematic features of the Martica-Boarezzo Fault; all the pictures are taken from the Ganna Granitic Stock, a) and b) hanging wall and footwall respectively high-angle reverse faults marked by fault steps and slickenlines; c) high-angle transpressive fault, associated with quartz veins and a thin quartz-porphyric dike, cut by a later high-angle normal fault (in blue); d) Left-lateral strike slip fault in the hanging wall of the MBT: note the associated R and P-fractures; e) Right-lateral strike slip fault : note the associated P-fractures and offset quartz veins.**



**Figure 6: a) middle Permian unconformity highlighted by the contact between the lower Permian Ganna Granitic Stock and the**
**Anisian Bellano Fm.; b) detail of the base of the Bellano Fm.; c, d, e, f, mesoscopic structural features of Valganna Fault (c, d, e, f).**

The Valganna Fault core is entirely developed in the Anisian–Ladinian succession showing fault breccia to fault gauge up to ca. 10 m thick (Figure 6f), surrounded by a tens of meters wide discrete damage zone. Fault slip data along the Valganna Fault (Figure 7a) point to an extensional to slightly transtensional kinematics. High-angle, SE-dipping fault surfaces are most
common along the ramp segment of the Valganna Fault, which features three sets of normal faults and strike-slip faults (Figure 7b). The slip inversion of these faults, after backfolding, indicates E-W extension (Figure 7c). Along the flat segment of the





Valganna Fault, two main sets of normal and transtensive faults (Figure 7a) indicate WNW-ESE directed extension, after backfolding (Figure 7b).



**Figure 7: Fault slip data and inversion: a) fault slip data along the faults described in the text (see Figure 3 for the location of the structural stations); b) kinematic inversion, P and T axes and the best fit fault plane solutions are represented; c) kinematic inversion after restoration to the nearest bedding orientation (S0); Codes: MBF: Martica-Boarezzo Fault; VGFr: Valganna Fault – ramp segment; VGFf – Valganna Fault – flat segment.**





**4.3 Thermochronology**

**4.3.1 (U-Th)/He dates**

Five samples, out of nine, from gneiss and micaschists provided good quality zircons for (U-Th)/He (ZHe) dating. Sample
mean ages and their one standard deviations (1s) are given in Table 1 and reported both in the geological map (Figure 3) and
in the geological cross-sections (Figure 4). Sample details are outlined in the Supplementary Table 1, presenting measured
quantities of U, Th, Sm, and He along with their uncertainties calculated from one standard deviation (1s) of the measurements.
Additionally, the table includes single grain ages and their corresponding propagated measurement uncertainties (1s). Three
samples (VA05, VA06, VA07) are in the hanging wall of the Marzio Fault close to the Ganna Granitic Stock, and two samples
(VA03 and VA08) are from the block that acted as footwall of the Mt. Nudo Fault and of the Marzio Fault (Figure 2). One
sample (VA07) includes 11 grains with a mean Late Triassic ZHe age of 226 ± 32 Ma (1s, 14%); three samples (VA05, 06,
and 08) include four to seven grains with mean ZHe ages in the range from 268 to 284 Ma, which is early Permian, and with
standard deviations (1s) varying from 19 Ma to 34 Ma (from 4% to 12%); finally, one sample (VA03) with four grains gives
a late Carboniferous age of 311 ± 23 Ma (1s, 7%).

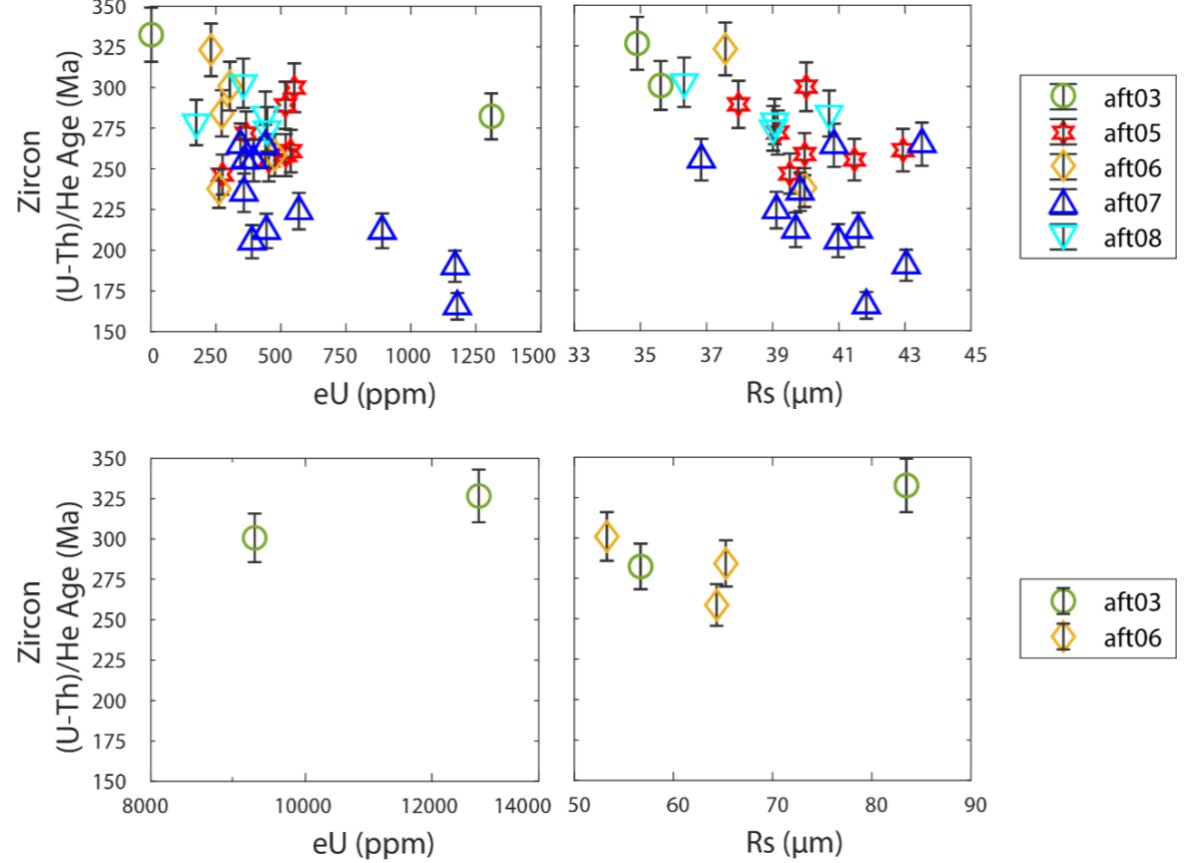



**Figure 8: Diagram showing all zircons (U-Th)/He data plotted against effective uranium (Ue) concentrations and equivalent spherical radius (Rs). Uncertainties are one standard deviations.**

The standard deviation of samples VA06 and VA07, 12% and 14%, respectively, is significantly higher than the 6% of the Fisch Canyon Tuff zircons, which were processed together with our samples. This observation suggests that samples VA06 and VA07 could be over-dispersed. Age dispersion can reflect either different zircon features like, for instance, differences in grain size, in radiation damage, and in U and Th distributions, or simply analytical uncertainties that cannot be accounted for. Zircon dimensions control the size of the He diffusion domain and this results in positive correlations among ZHe closure temperature, zircon dimensions, and ZHe ages. The radiation damage accumulated in zircons also controls the closure temperature and, in turn, is controlled by the zircon U and Th concentrations and thermal history (Shuster et al., 2006). The effect of the radiation damage on the ZHe closure temperature is non-linear, and it can be identified by plotting ZHe ages against the effective U concentration (eU), which is a proxy for the radiation damage (Gautheron et al., 2009). The U and Th distribution in zircon controls the alpha-ejection and if not accounted for, as in our case, can result in an inaccurate alpha-ejection correction and in age dispersion (Hourigan et al., 2005). All these effects can occur at the same time making it difficult to separate them. In our samples, we observe a slight negative age-eU relationship only for VA07 and no correlation with grain size for any sample (Figure 8). Thus, the age overdispersion in VA06 and VA07 could relate partly to the effect of radiation damage and partly to other effects that we cannot account for.

| Sample Code | Coordinates (Lat.; Long.) - WGS84 | Elevation (m a.s.l.) | N. grains | Age (Ma) | Standard deviation 1 s (Ma) |
|---|---|---|---|---|---|
| **VA03** | 45.956203°; 8.667739° | 205 | 4 | 310.53 | 23.33 |
| **VA05** | 45.911899°; 8.830069° | 563 | 7 | 268.82 | 19.32 |
| **VA06** | 45.892824°; 8.840748° | 796 | 5 | 280.92 | 33.73 |
| **VA07** | 45.929817°; 8.886418° | 358 | 11 | 225.74 | 32.38 |
| **VA08** | 45.956942°; 8.770584° | 340 | 4 | 284.76 | 12.53 |

**Table 1: Summary of (U-Th)/He age results; see Figure 1 for sample location on map and Figure 7 for locations in geological cross-sections.**

We used the Welch t-test (Welch, 1947) to test whether our samples have equal ZHe mean ages: results (Table 2) indicate that the Triassic sample is younger than the other samples; the three early Permian samples have equal ZHe ages and the late Carboniferous sample has equal mean as two out of three of the early Permian samples and could be slightly older than the third sample.



|  |  | Triassic | Permian | | | Carboniferous |
|---|---|---|---|---|---|---|
|  |  | VA07 | VA05 | VA06 | VA08 | VA03 |
| Triassic | VA07 | - | <0.05 | <0.05 | <0.05 | <0.05 |
| Permian | VA05 |  | - | 0.498 | 0.133 | <0.05 |
|  | VA06 |  |  | - | 0.823 | 0.165 |
|  | VA08 |  |  |  | - | 0.114 |
| Carboniferous | VA03 |  |  |  |  | - |

**Table 2: t-test on the obtained (U-Th)/He ages to check the probability that two samples derive from two overlapping distributions sharing the same mean; Triassic ages highlighted in yellow, Permian n blue and Late Variscan in red, for ease of comparison.**

In the study area, previous thermochronologic ages include a schist sample that has a 256 Ma old zircon fission-track age (Giger, 1991) and a Permian volcanoclastic sample with a ZHe cluster in the range between 288 Ma to 280 Ma and a younger ZHe age cluster between 82 and 74 Ma (Beltrando et al., 2015). The age clusters of this volcanoclastic sample record Permian cooling, no rift-related He-loss and possibly some early Alpine heating. This sample is close to one of ours, but we find no evidence of an Alpine thermal overprint in any of our samples that include 31 zircons in total. Thus, it is possible that the

Alpine thermal overprint is restricted locally. The complex ZHe age distribution of the volcanoclastic sample can also be interpreted speculatively as reflecting partly the clastic nature of the sampled rocks, which implies the possible presence of zircons with a very long geologic history and a complex response to low-T thermal events. A similar line of argument suggests that the limitedly dispersed and relatively simple ZHe age distributions of our samples could relate to the fact that our samples are from Carboniferous metamorphic rocks that may have been metamorphosed to temperatures high enough to anneal the

inherited history of the zircons. However, it is also possible that the relative tight age distribution of our samples reflects the small number of grains per sample, which may only partially sample the full ZHe age distribution. In any case, our samples bear no to modest post-Permian rejuvenation: three out of five of our samples have mean ZHe ages that are Permian and the remaining two samples have mean ZHe ages that are late Carboniferous and Late Triassic, respectively.

### 4.3.2 The effect of the granitic intrusion onto the thermochronologic record

The proximity of some of our samples to a granitic intrusion, i.e., the Ganna Granitic stock, which is early Permian in age, suggests that post-magmatic cooling rather than cooling related to exhumation could affect the ZHe ages. Three samples (VA05, VA06, and VA07) are very proximal (≤2 km away, Figure 3) to the Ganna granitic stock, whereas two samples (VA08 and VA03) are several kilometres far from the intrusion, about 5 km and >10 km, respectively. Among the samples proximal to the intrusion, only two (VA05 and VA06) have ZHe ages similar to the intrusion age whereas one sample (VA07) has a



Triassic ZHe age. The samples far from the intrusion (VA08 and VA03) have ZHe ages similar within uncertainty to the Ganna

intrusion age.

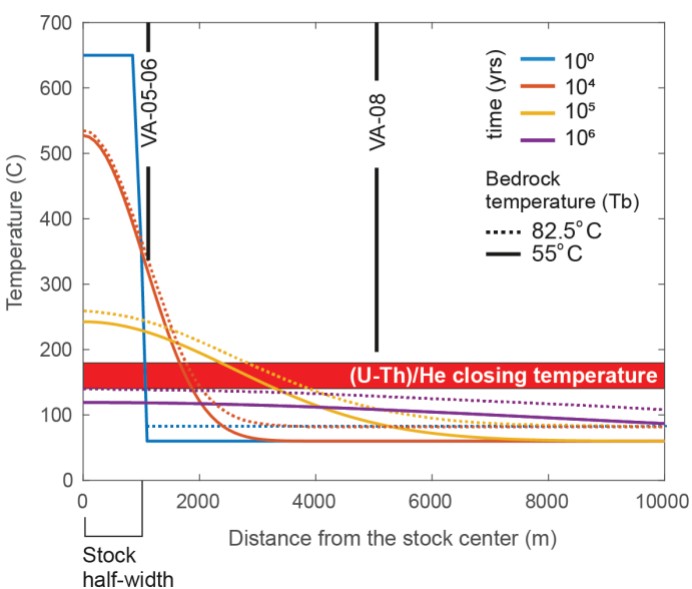

**Figure 9: One dimensional transient thermal response to an intrusion (halfwidth = 1000 m). Intrusion depth was assumed to be shallow and into country rock at background temperature ranging between 55oC and 82.5oC.; initial intrusion temperature of 650**

**°C, and thermal diffusivity of 32 km2/Myr.**

In order to test for the thermal effect related to the intrusion, we used a 1-D heat-transfer model at shallow depth combined with an estimate of the time-temperature conditions, at which a heating event could cause He loss in zircons. For the heat-transfer model, we assumed a background temperature ranging between 55 and 82.5 °C, consistent with a near-surface intrusion

depth (i.e., between 2 and 3 km of maximum depth of emplacement, based on the thickness of the overlying units), an initial intrusion temperature of 650 °C and a thermal diffusivity of 32 km$^2$/Myr (Figure 9). These inputs result in temperatures close to the intrusion temperature for 1 yr at a horizontal distance of 1 km from the granite and temperatures in the range of 200 °C and to 250 °C for 0.1 Ma at 2 km distance. To calculate how long a zircon should be held at a certain temperature to cause 90% He loss, which would cause age resetting, we used the partial loss-only He diffusion equations (Reiners and Brandon,

2006). In these equations we input average ZHe kinetic parameters (Reiners et al., 2004) and grain dimensions equal to the mean equivalent spherical radius (Rs; Supplementary Material Table S1) of the zircons dated in this study (Rs: 45 µm). We obtained that a steady temperature close to 400 °C for 1 yr results in 90% He loss whereas a steady temperature close to 200 °C requires at least 1 Ma time to produce 90% He loss. Thus, the early Permian ZHe ages of samples VA08 and VA03, which are more than 5 km away from the intrusion, likely record cooling related to exhumation whereas the early Permian ZHe ages





of samples VA05 and VA06, which are proximal to the intrusion, likely reflect post-magmatic cooling. Finally, the Triassic
ZHe ages of sample VA07 is too young to reflect post-magmatic cooling and suggest a protracted thermal history.

## 5 Discussion

### 5.1 The Variscan-Alpine Cycle transition

The data presented here provide new temporal and kinematic constraints for the events that occurred during the transition
between the Variscan and Alpine cycles, constraining the age of the earliest crustal rifting stage, at the inception of the Alpine
cycle, and defining its temporal and kinematic relations to earlier tectonic phases.
The tectonic and thermochronological datasets demonstrate that, in the study area, the Variscan metamorphic basement was
exhumed during the early Permian, flattened by a regional-scale erosive phase, and then dissected into a continental endorheic
basin, bounded, to the north, by the Marzio Fault. While this structurally-controlled basin was progressively filled with calc-
alkaline volcanic-volcanoclastic products, the upper crust was intruded at shallow depths by a sub-volcanic body, i.e., the
Ganna Granitic Stock. The tectonic control exerted by the Marzio Fault is testified by: i) the differential subsidence of the
southern block compensated by the accumulation of a thicker extrusive sequence, and ii) the structurural control on the
emplacement of the Ganna Granitic Stock. Thermochronological results, in fact, combined with thermal diffusion modelling,
exclude that the stock could be present also in the northern block, below the exposed basement. Additionally, since the early
Permian, the basement rocks of the study area have been relatively stable at near surface levels, at depths shallow enough that
most of them were not affected by any later thermal event. Two of our Permian ZHe ages (VA05: 269±19 Ma; VA06: 281±33
Ma) are from basement rocks south of the Marzio Fault that were reset by the Ganna Granitic Stock and bear no record of any
later thermal event. The fact that these two samples likely record post-magmatic cooling indicates that the Ganna Granitic
Stock intruded Carboniferous metamorphic rocks at 281 Ma, which by the early-middle Permian were already exhumed to
shallow crustal levels, at depths near or shallower than the Zhe closure temperature (i.e., ~180 ℃; Reiners and Brandon, 2006).
North of the Marzio fault, two samples have mean Zhe ages that are Permian (VA08: 285±12 Ma) and late Carboniferous
(VA03: 311±23 Ma), respectively. These two samples have statistically similar ZHe ages (T-test, Table 2), and therefore,
collectively, they indicate that also the basement block, north of the Marzio Fault, has been at shallow crustal levels since the
early Permian. Thus, four out of five of samples indicate exhumation at near-surface levels either during or before the early-
middle Permian. Our observations are consistent with previous studies and altogether they indicate that the early Permian
extensional phase undoubtedly occurred after the end of the Variscan cycle throughout the peri-Variscan terrain (Figure 10;
e.g., Ziegler and Stampfli, 2001, Stampfli and Kozur, 2006; Cassinis et al., 2012; Cassinis et al., 2018; Ballèvre et al., 2020).
Some Authors interpret this massive cal-alkaline magmatism as associated with the extension related to the back-arc opening
of the Paleotethys subduction zone (e.g., Visonà, 1982, Lorenz and Nicholls, 1984; Stille and Buletti, 1987; Di Battistini et al.,




1988; Finger and Steyrer, 1990, 1991; Bonin et al., 1993; Doglioni, 1995; Ziegler and Stampfli, 2001, Stampfli and Kozur, 2006; Karakas et al., 2019; Ballèvre et al., 2020).

No reliable fault kinematic indicators are available in the study area to further characterize the stress orientation of the early Permian tectonic phase. Notably, the most reliable observations and datasets nearby, in this line, indicate that pure-extensional detachment faults and transtensional structures come with crustal thinning and high temperature-low pressure metamorphism

(e.g., Phol et al., 2018; Roda et al., 2019; Zanchi et al., 2019; Festa et al., 2022; Locchi et al., 2022).

While a discussion regarding the geodynamic nature of the early Permian extensional phase may be somewhat speculative based on our data, our observations unequivocally show a lack of continuity between the early Permian rifting and the Mesozoic Alpine Tethys rifting (Figure 10). The early-middle Permian geologic record in our study area attests to the shifting from a rifting phase to a transpressional phase and to the development of the middle Permian unconformity. Indeed, the activity

of the Martica-Boarezzo fault and Mondonico push-up is consistent with the observation by Cadel et al. (1986) and Gretter et al. (2013) of folded lower Permian deposits in the central Southern Alps suggesting a middle Permian compressional tectonic phase, and with paleomagnetic (Muttoni et al. 2003, 2009) and stratigraphic data (Cassinis et al., 2012; Gretter et al., 2013; Cassinis et al., 2018;). To our knowledge, the Martica-Boarezzo fault and Mondonico push-up represent the first direct observation of structures related to the middle Permian compressional phase testifying to a regional-scale significance of this

event. Some studies suggested that at this time there was indeed a change in the plate configuration from Pangea A to Pangea B between Gondwana and Laurasia (Muttoni et al. 2003, 2009, and references therein). While other studies suggested that transpression is related to a dextral megashear system that was active during the oblique subduction of the Paleotethys and the opening of the Neotethys ocean (e.g., Cassinis et al., 2012; Gretter et al., 2013; Cassinis et al., 2018; see Torsvik and Cocks, 2004 for a complete discussion). However, the discussion supported by our dataset does not favour either of the two

hypotheses.

## 5.2 The Alpine Cycle inception

The end of the middle Permian transpressional phase is marked by the onlap of the late Permian-Early Triassic siliciclastic wedge related to the marine ingression of the Paleo-Tethys from the east (Figure 10; e.g., Bernoulli, 2007) and it is followed by a renewed phase of extension that evolved through several stages.

During the first stage, in our study area, the Middle Triassic tectonostratigraphic data constrain a significant increase in subsidence that allowed the accommodation of up 800 m thick platform carbonate alternating with intraplatform basinal sediments. Subsidence was associated with E-W oriented extension that was controlled by the activity of the Valganna Fault and that resulted into the dissection of the depocenter into sub-basins. The brittle nature of the ramp and flat segments of the Valganna fault suggests that extension at this time was distributed within the shallower zone of the upper crust.



The Middle Triassic extensional phase has been interpreted as related to the far field effect of the Meliata-Maliac oceans (e.g., Castellarin et al., 1980; 1988; Ziegler and Stampfli; 2001; Stampfli et al.; 2002; Armienti et al., 2003; Zanetti et al., 2013, Beltrando et al., 2015) or as the onset of the Alpine-Tethys opening (e.g., De Min et al., 2020; Real et al., 2023).

Finally, in the second stage, during the Late Triassic extension was associated to localized lithospheric normal faulting that ultimately led to crustal thinning and break-up (e.g., Bertotti et al., 1993; Gaetani et al., 2010).

In the Late Triassic-early Jurassic, extension continued with an E-W direction, but localized onto few master faults located outside our study area, both to the west (Mt. Nudo Basin; Figure 1) and to the east (Lombardian Basin; e.g., Bernoulli, 1964; Bertotti et al., 1993), where kilometre-thick syn-rift sequences are preserved.

To the east of our study area, the Late Triassic-Jurassic crustal thinning produced a widespread thermal anomaly that affected the thermochronologic record of the basement rocks of the central Southern Alps and emplacement of ore deposits (Bertotti et 460 al., 1999; Giorno et al., 2022). In our study area, the thermal overprint is evident in only one of our samples (VA07), situated in the easternmost region, at the lowest elevation, and within the deepest layers of the tectonic-stratigraphic sequence. This sample has a mean age dating to the Late Triassic (226±32) and has over-dispersed grain ages between 260 and 170 Ma in a negative relationship with eU (Figure 8). This suggests that this sample accumulated high radiation damage over a long residence at shallow crustal levels and that it was affected by a partial thermal rejuvenation that possibly started in the Triassic 465 and continued until the early Jurassic.

Thus, altogether our data support the idea that the Ladinian-Carnian extensional phase predates crustal thinning that started in the Norian (e.g., Figure 10; Bertotti et al., 1993; Gaetani et al., 2010). In this view, deformation in the Ladinian-Carnian, was rather distributed resulting in modest amount of extension concentrated within the upper crust (Figure 10; Mohn et al., 2010; 2011; 2012; Péron-Pinvidic et al., 2013; Naibof et al., 2017) whereas from the Norian extension localized along few 470 lithospheric detachment faults and led to crustal thinning (Figure 10; Péron-Pinvidic et al., 2013; Naibof et al., 2017). The co-axiality that we observed between the extensional events in the Ladinian-Carnian and the following ones supports the idea of a continuity of the geodynamic setting from the Middle Triassic through the Jurassic. This inference is concordant with the evolution of extension as proposed by previous studies in the central Southern Alps (e.g., Bertotti et al., 1993; Gaetani et al., 2010). This view is also supported by the geochemical data from magmatic rock in the Dolomites indicating that the Ladinian-475 Carnian magmatism in the eastern Southern Alps was related to the lithosphere thinning and mantle upwelling due to rifting events that ultimately caused the break-up in the Late Triassic-Early Jurassic (De Min et al., 2020).







**Figure 10. Conceptual and summary scheme of the Variscan-Alpine Cycle transition as constrained in this work and from other works (WSA, Western Southern Alps; CSA, Central Southern Alps): chronostratigraphic scheme redrawn after Schalteger and Brack, 2007; Berra et al., 2009; Cassinis et al., 2012; Gretter et al., 2014; Beltrando et al., 2015; Cassinis et al., 2018; sections of the former Southern Alps redrawn after Beltrando et al., 2015.**





## 7 Conclusions

This study provides new constraints on the Variscan-alpine cycle transition, a period relatively poorly documented within the
European Southern Alps. In particular, our data shed light on the transition from the early Permian rifting to the first inception
of Triassic crustal extension, through a phase of transpression, previously undocumented in the study area.

In the framework of a regional perspective, the main findings of this work are:

- During the early Permian, a first rifting stage resulted in the accumulation of a thick pile of volcanic and volcano-
clastic successions, deposited in a structurally controlled endorheic basin. At the same time, the emplacement of a
fault-bounded intrusive stock at shallow depths caused the sudden heating of the surrounding basement rocks,
followed by a later cooling at shallow crustal levels where they resided for the rest of their time.
- The sudden cessation of magmatic activity, followed by transpressive tectonics (early p.p. – middle Permian) and a
later regional-scale erosion (middle Permian), marks a distinct discontinuity in the regional Permo-Triassic
evolution, that is the conclusion of a first cycle of crustal rifting and the shift toward the successive Alpine cycle.
- At the Middle Triassic, the inception of a second rifting event, here documented along a well-exposed normal fault,
resulted in the exhumation of the fault footwall, as recorded by thermo-chronological dates, the subsidence of the
fault hanging wall with drowning of the carbonate platform and development of a fault-bounded anoxic basin.
- It is suggested that the Middle Triassic extension could represent the stretching phase related to the onset of the
Alpine Tethys rifting.

The main implications of our findings are that the onset of the Alpine cycle dates to the Middle Triassic and that there is no
continuous Permo-Triassic extension, as previously suggested by other Authors (e.g., Winter and Bosellini, 1981), but there
are multiple and distinct stages of rifting.

## Appendices

### Data availability

Fault slip data inversion were calculated using FaultKin v.8 software (https://www.rickallmendinger.net/faultkin last accessed
03/04/2024). Thickness analyses were performed using the MOVE software (IPM v13.0 software suite, courtesy of Petroleum
Experts Ltd). (U–Th)/He (ZHe) analysis results are available in the supplementary material Table S1 of this article. Geological
map was done with QGIS 3.28.11 (https://qgis.org/it/site/ Last accessed 03/04/2024) and it is shown in Figure 3 of this article.



**Author contribution**

Conceptualization: ES, FL, GF; Fieldwork and sample collection: ES, FL, Data collection: GF, CM; Data analysis: ES, FL,
GF, CM; Methodology: ES, FL, GF, CM; Writing original draft: ES, FL, GF; Writing, review, and editing: ES, FL, GF, CM.
All authors read and approved the final manuscript.

**Competing interests**

The authors declare that they have no conflict of interest.

**Funding**

This work benefited from funds from the PhD project of Emanuele Scaramuzzo.

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
