# Peer review of "Evidence for Multi-Rifting in the Variscan-Alpine Cycle Transition: Insights from the European western Southern Alps"

_EGUsphere, 2024_

## Referee Comment (RC2)

[referee-annotated manuscript omitted]

---

## Author Response (AR1)

COMMUNITY COMMENT 1 (H. Seebeck)

COMMENT: Given how much weight thickness variations in the early Permian effusive units and overlying Anisian – Ladinian sedimentary strata have in determining fault activity, I am a little surprised at how little information is provided in the methodology about how thickness estimates were derived. The author states in Line164 That "the geometry and orientation of geological structures were extrapolated in three-dimensions starting from a series of cross-sections that were integrated with constraints from both the surface trace of planar features and direct field measurements." This is followed by "The lower Permian succession records the activity of the Marzio Fault." Line225.
REPLY: In the method section we described how we derived the isopach map from a 3D model. This model is based on 25 cross-sections and 4 stratigraphic sections, whose thickness was measured in the field. We will add more specific information in the method section to emphasize how data for the thickness analysis were collected, and in the Supplementary material we will enclose a move file with the full restoration procedure and the resultant 3D mesh surfaces. Our model and relating thickness-analysis is referred to the Anisian-Ladinian succession only, lines 163-163: "From the thickness-analysis, we derived an isopach 2D map contouring at the top of the Middle Triassic."
We will edit the text in line 225 as it is indeed confusing.
ACTUAL CHANGES TO THE TEXT: We have revised section 3.1 thoroughly by adding the details of the construction of the 3D model and of the isopach map. In the supplementary information (Supplementary S1), we added the mesh surfaces of the 3D model as 3D pdf file. We modified Figure 3 to show the locations where we measured stratigraphic sections. We modified Figure 4 by adding a map of the grid of cross-sections that we used to build the 3D model.

COMMENT: By the nature of the description of the effusive units comprising tuffs, ignimbrites and intermediate and felsic lavas, the extrapolation of thicknesses becomes problematic given the highly variable modes of emplacement. Thickness variations of hundreds of metres over a few kilometres could be expected depending on source vent location and the height of pre-existing topography. I would be cautious on making any interpretation of fault activity based on thickness variations of eruptive units, particularly those associated with ignimbrites and pyroclastic flows without a good understanding where these units originated from. For example, if you are close to the boundary of a caldera then thickness variations say nothing about tectonic fault activity.
REPLY: We will expand the discussion to account for the possible role of inherited volcanic morphology and to present critically our interpretations of the Valganna volcanic sequence.

While thickness variations have indeed been used to infer fault activity, this has been done especially for the Middle Triassic extensional phase. Our 3D model does not extend to the lower Permian succession.

We acknowledge that thickness variations within the Permian succession may be influenced more by the geometry of volcanic bodies and pre-existing topography than by fault activity. Nevertheless, we maintain that the Marzio Fault played a role, supported by regional observations and the apparent bounding of the Ganna Granitic Stock. In fact, the Marzio Fault has been interpreted as active during the Late Carboniferous (Casati, 1978), particularly in relation to the distribution of the Carboniferous strata, such as the Manno Conglomerate.

We propose that renewed activity along the Ganna Fault during the Permian may have controlled the emplacement of the Ganna Granitic Stock. We will elaborate on these points in the introduction by reviewing previous studies and in the discussion, where we will also explore alternative interpretations.

ACTUAL CHANGES TO THE TEXT: We modified the discussion in section 5.2. There we wrote: "While this structurally-controlled basin was progressively filled with calc-alkaline volcanic-volcanoclastic products, the upper crust was intruded at shallow depths by a sub-volcanic body, i.e., the Ganna Granitic Stock. The geometry of the volcanic deposits during this phase appears to be influenced by the tectonic control exerted by the Marzio Fault, as suggested by the apparent differential subsidence of the southern block, which is associated with the accumulation of a thicker volcanic succession. However, it is important to note that these variations could also reflect lateral changes in the thickness of the volcanic beds, the localized presence of lava domes, and the paleotopographic setting during the emplacement of the volcanic units.

Additional observations support the significant structural influence of the Marzio Fault on the volcanic and magmatic activity during the Middle Permian. Thermochronologic data suggest exhumation of the Marzio Fault's footwall before and during the volcanic activity, aligning with its normal fault regime. Field observations from this and previous studies (Bernoulli, 2018; Govi, 1960; Bakos et al., 1990) further support this interpretation, as the Ganna Granitic Stock and associated dikes, which may intrude the overlying basement unit, are absent north of the Marzio Fault."

COMMENT: How many cross-sections were used and where were they located? Only four are shown in the manuscript.

REPLY: Our 3D geological model is based on 25 cross-sections. We will add a map with the traces of the cross-sections in the method sections and the meshes of the surfaces used for the restoration in the Supplementary Materials.

ACTUAL CHANGES TO THE TEXT: We modified Figure 4 and the Methods section. The revised Figure 4 shows the grid of cross-section used to build the 3D model. In section 3.1, we wrote: "We built 25 geological cross-sections (locations in Figure 4c) that informed a3D geologic model, developed using the MOVE software (courtesy of Petroleum Experts Ltd). The model is composed of mesh surfaces representing the top horizons of stratigraphic units and fault surfaces. These mesh surfaces are available as 3D pdf file in supplementary material (S1)."

COMMENT: I note some inconsistencies with the cross-sections presented and the geologic map.
REPLY: In sections A and B, we will fix the dip angle of the base of the volcanic sequence and some additional minor issues. We underline that since we have not restored the volcanic series this does not affect the thickness analysis.
ACTUAL CHANGES TO THE TEXT: we modified figure 4, section A and B.

COMMENT: For example the dip of the Anisian-Ladinian units on the northwestern side of the Marzio Fault dip to the northwest at 45° however the section shows a very shallow southeast dip.
REPLY: The succession cropping out on the northwestern side of the Marzio Fault is involved into a series of detached folds (see sections in figure 4). The northwest dipping panels refer to the northern limb of the anticlines (see for example the northern limb of the Mt. Pian Nave fold) and to the southern limb of the synclines.

COMMENT: The thickness of the Effusive unit in section A southeast of the Marzio Fault is difficult to understand as the anticline shown in the map does not appear to have been represented?
REPLY: The southern block of the Marzio Fault is fully involved into a north-verging anticline with a periclinal closure to the west (Figure 3). This anticline, also referred as Arbostora or Maroggia-Brinzio Anticline, has been described also in previous studies (Kälin & Trümpy; 1977). This structure has been interpreted as a deeply rooted fault-propagation fold and we analyzed it in a previous work (Scaramuzzo et al., 2022). To the east, the fault-related fold is presently eroded and the offset rapidly increases eastward giving place to a break-through fault-propagation fold, whose front limb is eroded. Close to the fault trace, only the backlimb of the fold is fully preserved. We will add the projection of the anticline in all the sections.
Kälin O. & Trümpy D.M., 1977: Sedimentation und Paläotektonik in den westlichen Sudalpen: Zur triasisch-jurassischen Geschichte des Monte Nudo-Beckens. Ecl. Geol. Helv. 70/2, 295-350.
ACTUAL CHANGES TO THE TEXT: Figure 4 was modified by adding the projection of the anticline on the sections.

COMMENT: There appears to be a c. 20° decrease in the dip between the underlying and overlying units despite being shown to have a parallel contact geometry?
REPLY: The graphical rendering of the sections can be misleading. There is indeed an angular unconformity between the lower Permian and the Triassic. The contact between the Permian and the Triassic series is represented by the Middle Permian Unconformity that is parallel to the bedding of the overlying Middle Triassic.
ACTUAL CHANGES TO THE TEXT: we modified figure 4, section A and B.

COMMENT: Where is the MNF Fault shown on the cross-section in Figure 3? Where are the structural stations on Figure 3?
REPLY: We will add the label MNF, the structural stations and the cross-sections in figure 3.
ACTUAL CHANGES TO THE TEXT: we modified Figure 3 by adding the structural stations and the label MNF.

COMMENT: While these are relatively minor points I find these inconsistencies reduce my confidence in the structural model and the interpretations that follow.
REPLY: In the revised manuscript, we will show in greater detail how the 3D model has been built, how many cross-sections have been drawn, and we will discuss at depth the hard constraints to the extrapolated geometries and the weak constraints, given the assumptions for the extrapolation of surfaces. We hope that this will better illustrate the internal consistency of the model.
ACTUAL CHANGES TO THE TEXT: We revised the method section, section 3.1, Figure 1 to 4 to show with greater detail how the structural model was built and to amend some apparent inconsistencies in the cross-section shown in figure 4.

COMMENT: Thermochronology, while this is largely outside of my expertise I have a couple of comments about the interpretation of the age data.
Line390 states that the young age of VA07 indicates a protracted thermal history while VA05 and VA06 reflect post-magmatic cooling of the Ganna Granitic Stock. It appears to me from Figure 3 that all three samples are located within a few hundred meters of the contact between the metamorphic units and the intrusion and within a 2.5 km radius of one another.
REPLY: All three samples are located near the contact with the intrusion, as discussed earlier in this section (lines 366-367). The cooling ages of samples VA05 and VA06 closely match the emplacement age of the Ganna granitic stock. In contrast, the cooling ages of sample VA07 span a broader range, from 250 to

170 Ma, overlapping the Permian emplacement of the Ganna intrusion and extending into the Middle Jurassic. Our modeling results show that temperatures near a shallowly emplaced intrusion drop rapidly, both spatially and temporally. All three samples (VA05, VA06 and VA07) were close enough to the intrusion to be affected by its thermal effect. Yet, only the cooling ages of VA05 and VA06 reflect post-magmatic cooling, whereas the extended cooling period observed in VA07 is inconsistent with post-magmatic cooling only. The cooling record of sample VA07 reflects its lower structural position relative to samples VA05 and VA06 and to the Ganna Stock.

We will revise this section to clarify our conclusions.

ACTUAL CHANGES TO THE TEXT: We added a new section in the discussion with title: Permian thermal record. We moved part of the text that was in section 4.3.1. to this new section and we added a more in depth discussion of all the thermochronologic data.

COMMENT: Also how do you explain that VA08 and VA06 have essentially the same age, VA06 being a product of post-magmatic cooling while VA08 is not? Line403 states that thermal diffusion modelling excludes the presence of the Ganna Granitic Stock below the exposed basement. The author therefore needs to explain why VA08 has an identical age to VA06 but was not part of the thermal event that generated identical ages.

REPLY: This topic is addressed in section 5.1, which we will revise to provide a more detailed interpretation. Samples VA08 and VA06 exhibit similar cooling ages, overlapping with the emplacement age of the Ganna granitic stock, due to the combined influences of intrusive emplacement and normal activity of the Marzio fault. The thermal effect of the intrusion in the hanging wall of the Marzio fault caused sample VA06 to record post-magmatic cooling. Meanwhile, the normal faulting activity led to the uplift and exhumation of the footwall, recorded by sample VA08. Consequently, the typical cooling pattern across a normal fault—where younger cooling ages are found in the footwall—has been overprinted by the thermal impact of the granitic intrusion.

ACTUAL CHANGES TO THE TEXT: In the new discussion section with title " Permian thermal record",  we discuss why there is no apparent thermochronologic offset across the Marzio Fault and, yet, our data are consistent with the normal activity of this fault during the early-middle Permian. The new text includes the following discussion: "Despite no thermochronologic age offset across the fault, our data are consistent with the normal activity of the Marzio fault during the early-middle Permian. The lack of offset likely reflects the thermal overprint from the granitic intrusion, which masked the typical cooling age pattern across a normal fault, where younger cooling ages would be normally observed in the footwall (e.g., Willett et al., 2021). In this context, VA08 and VA03 record exhumation of the footwall prior to and during the

emplacement of the Ganna Granitic stock, while VA05, VA06 and VA07 were thermally reset by the intrusion. The slightly older Carboniferous age of VA03 may reflect its greater distance from the Marzio Fault, consistent with the trend of younger ages in footwall rocks closer to a fault. The younger cooling age of VA07 may relate to its lower elevation in the tectonic-stratigraphic succession of the study area, near the eastern margin."

COMMENT: Unless the Ganna Granitic Stock has a protracted emplacement period spanning at least 50 Myrs, I find it difficult to reconcile how these samples have had a significantly different thermal history. Figure 1 and 10 shows intrusions in the same age range further to the west but this is not really discussed? Could the sample locations have come from different depths with VA07 recording more exhumation? What other reasons could there be for the differences? Where on Figure 2 are your samples from, at least show the study area on this figure.

REPLY: It is unlikely that the Ganna Granitic stock experienced a protracted emplacement. Sample VA07 is in a position structurally lower than the other samples in the hanging wall of the Marzio Fault – this difference is reflected by the cooling record of this sample. This is clearly explained in the lines 458 to 465. We will expand the discussion in these lines to provide further clarity.

ACTUAL CHANGES TO THE TEXT: in the new discussion section about the thermochronologic data, we wrote: "The younger cooling age of VA07 may relate to its lower elevation in the tectonic-stratigraphic succession of the study area, near the eastern margin. This margin is near the area affected by Late Triassic-Jurassic crustal thinning, generated by a widespread thermal anomaly that affected the thermochronologic record of the basement rocks of the central Southern Alps and resulting also in the emplacement of ore deposits (Bertotti et al., 1999; Giorno et al., 2022). The ZHe age distribution of VA07, along with its negative relationship with eU (Figure 8) suggests that this sample accumulated high radiation damage over a long residence at shallow crustal depths, with partial thermal rejuvenation possibly starting in the Triassic and persisting into the early Jurassic. Consequently, the Late Triassic-Jurassic thermal anomaly in the Southern Alps is evident in only one in VA07."

COMMENT: These seeming inconsistencies in the interpretation of the thermochronology lead me to question either the validity of the thermal diffusion modelling or the suggestion that there aren't granites beneath the northern block.

REPLY: The modeling we present is simple and straightforward, designed to quantify the thermal effects of a shallowly emplaced granitic intrusion. Our interpretation of the thermochronologic data is primarily based on the data itself and its relationship to field-observed structures. While we do not believe

there are significant inconsistencies in our interpretation, we acknowledge the need for greater clarity in our discussion of the data and will work to address this.
ACTUAL CHANGES TO THE TEXT: We added a new section in the discussion where we revisited our discussion of the thermochronologic data. We refer the reader to the new section 5.1.

COMMENT: If I look at Figure 1 I can see Early Permian granite to the east of the study area that would be consistent with intrusions underlying the northern block.
REPLY: Here there is a misinterpretation of the regional structure of the Southern Alps. There are several Permian intrusive and effusive districts in the Southern as shown in the map of Figure 1. Hower there is no physical correlation between these districts and the Ganna Granitic stock, which belongs to the Lugano-Varese district.

COMMENT: Further to this, I do not feel the author has demonstrated structural control on the emplacement of the Ganna Granitic Stock with the observations presented here. Faults will tend to localize in regions where there are strength contrasts so the author would need to either demonstrate that the fault pre-dated the intrusions or that there are structural or cooling fabrics consistent with fault motion during the time of emplacement.
Otherwise faulting may have just exploited strength contrasts associated with the intrusion. This also relates to whether intrusion has occurred beneath the surface in the northern block.
If you cannot rule out that VA08 was associated with the thermal event associated with intrusion then fault control for the emplacement of the intrusion is less likely.
REPLY: The Marzio fault was indeed active already in the late Carboniferous and therefore predate the intrusion (Casati, 1978). During the Ganna Granitic Stock emplacement was reactivated as a normal fault.
We will add this observation in the discussion to highlight that the Marzio Fault predated the granite emplacement.
ACTUAL CHANGES TO THE TEXT:
In the revised text, in section 5.1, concerning sample VA08, we wrote; "North of the Marzio fault, two samples show Permian (VA08: 285±12 Ma) and late Carboniferous (VA03: 311±23 Ma) ZHe ages. Despite being statistically similar (T-test, Table 2), VA08 aligns with the timing of the Ganna Granitic Stock intrusion. A 1D thermal model indicates that the thermal effect of a shallowly emplaced granitic intrusion drops rapidly both in space and time, and no field evidence suggests a granitic intrusion near VA08. While buried, unexposed granitic

intrusions near VA08 are possible, given the landscape's topography, with incised valleys, the likelihood of unexposed granites is minimal."

In section 5.2 of the revised text, we wrote:" The geometry of the volcanic deposits during this phase appears to be influenced by the tectonic control exerted by the Marzio Fault, as suggested by the apparent differential subsidence of the southern block, which is associated with the accumulation of a thicker volcanic succession. However, it is important to note that these variations could also reflect lateral changes in the thickness of the volcanic beds, the localized presence of lava domes, and the paleotopographic setting during the emplacement of the volcanic units.

Additional observations support the significant structural influence of the Marzio Fault on the volcanic and magmatic activity during the Middle Permian. Thermochronologic data suggest exhumation of the Marzio Fault's footwall before and during the volcanic activity, aligning with its normal fault regime. Field observations from this and previous studies (Bernoulli, 2018; Govi, 1960; Bakos et al., 1990) further support this interpretation, as the Ganna Granitic Stock and associated dikes, which may intrude the overlying basement unit, are absent north of the Marzio Fault."

MINOR COMMENTS

Line22: "architectural framework" – what does this mean? Vague term, do you mean structural framework?
REPLY: We will change it accordingly.
ACTUAL CHANGES TO THE TEXT: We replaced architectural with structural.

Line23: "Nonetheless" doesn't seem appropriate here. I would reverse this sentence by stating that "Due to the large hiatus in the geological record...the transition between the two-cycles is open to different interpretations."
REPLY: We will change it accordingly.
ACTUAL CHANGES TO THE TEXT: we changed as suggested.

Line56: Southalpine one word?
REPLY: It can be used both ways.

Line58: phase not phases?
REPLY: We will change it accordingly.
ACTUAL CHANGES TO THE TEXT: we changed as suggested.

Line86: Please make the description of the Early Permian volcanic units consistent throughout the text and figures. Called Volcanic units in Figure 1, Effusive units in Figure 3 and 4 but not described as such in the text (or least not explicitly).

REPLY: We will change it accordingly.
ACTUAL CHANGES TO THE TEXT: we changed as suggested.

Line118: Como Lake is mentioned in the text a couple of times but not shown on a map? Lake Como or Como Lake? Consistency through text.
REPLY: The Como Lake is shown in figure 1 but it is not labelled. We will add the label.
Line134: Lake Maggiore not shown on map.
REPLY: It is shown in figure 1 but the label is missing. We will add it.
ACTUAL CHANGES TO THE TEXT: we added the label in Figure 1.

Line170: "key horizons that were assumed as horizontal…" This is a large assumption in a volcanic environment. Provide justification for this assumption.
REPLY: No horizons within the volcanic pile have been considered for restoration. Instead, we used the middle Permian unconformity, that is a planar erosive surface, as a key horizon.
ACTUAL CHANGES TO THE TEXT: the revised method section 3.1 should now clarify this point.

Line178: Add reference to support correspondence between stress and strain – in my opinion this is not straight forward particularly in transpressive or trantensional settings.
REPLY: Yes, we agree. The phrase in this line is confusing. The quantitative inversion of fault slip data provides direct constraints on the orientations and relative magnitudes of the global principal strain rates (e.g., Twiss and Unruh, 1998). Indeed, that is the reason why we used a kinematic approach and strain axes are reported in all the figures and in the text.
We will remove the sentence.
Twiss, R. J., & Unruh, J. R. (1998). Analysis of fault slip inversions: Do they constrain stress or strain rate?. Journal of Geophysical Research: Solid Earth, 103(B6), 12205-12222.
ACTUAL CHANGES TO THE TEXT: in section 3.2, we specify that we calculated paleo-strain orientations.

Line200: Your samples are 10x older than the standard used, how might this affect the results?
REPLY: The Fish Canyon Tuff serves as an age standard, but it is not used to calibrate the measurements or the ages, and therefore it does not affect the results. As explained in the methods section, the age standard is processed alongside the samples to ensure the accuracy of the date estimates and to monitor intrasample dispersion. We also provided a summary of the

measurement procedures. Helium-4 ($^4$He) is measured using an external standard, an aliquot of $^4$He gas from a calibration bottle. Uranium and thorium are measured using internal standard solutions of $^{233}$U and $^{230}$Th, which are weighed and added to the sample before dissolution. We will add some additional specifications in the method.

ACTUAL CHANGES TO THE TEXT: the revised text specifies the following: "To verify the accuracy of date estimates and monitor intrasample dispersion, six zircons from the Fish Canyon Tuff were processed alongside the samples. These zircons were not used for calibration purposes."

Line227: "lower Permian" as per Line86 comment please make the description of the Early Permian effusive units consistent throughout the text, makes it difficult for the reader when you keep changing the way the same units are referred to.
REPLY: We will change it accordingly.
ACTUAL CHANGES TO THE TEXT: we replaced throughout the text and figures the term effusive with volcanic and volcanoclastic.

Line228: Section A is not the best example to use here as the thickness of the effusive units are not constrained on this section, it is only interpolated. Section C is constrained though refer to my earlier comment regarding dip angles, the dip on the effusive units appears c. 20° higher than the overlying unit which implies deformation prior to the deposition of the overlying sedimentary units.
REPLY: The thickness variation of the early Permian effusive units was also measured in the stratigraphic sections shown in the stratigraphic block diagram. This will appear clearer as we will add the map showing the location of the measured stratigraphic sections and the traces of the cross-sections.
ACTUAL CHANGES TO THE TEXT: we modified figure 1 and figure 4. Figure 1 shows the locations where stratigraphic successions were measured and figure 4 shows the grid of geologic section used for the 3D model.

Line235: "Middle Permian Unconformity seals the Martica-Boarezzo Fault". Truncates may be a better term, seal implies a fluid-flow property.
REPLY: We will change it accordingly.
ACTUAL CHANGES TO THE TEXT: we changed as suggested.

Line236: How is the Mondonico push-up kinematically compatible with the Martica-Boarezzo Fault? These faults are near at right-angles to one another? Describe how these structures are kinematically related.
REPLY: The thrusts delimiting the Mondonico push-up are perpendicular to the shortening direction. The Martica Boarezzo Fault act as a strike-slip fault oblique relatively to the shortening direction. Thus, the orientation and kinematic of

these faults is consistent with the same strain ellipsoid – the strain ellipsoid that we derived for the Martica Boarezzo Fault is shown in Figure 7.

We will describe this relationship in the result section.

ACTUAL CHANGES TO THE TEXT: as we revised the manuscript, we realized that our interpretation of the relationship between the Martica-Boarezzo fault and the Mondonico push-up was wrong. In the revised text, we describe the Mondonico push-up as a ridge bounded by high-angle NNW-SSE trending faults that appear possibly displaced by the Martica-Boarezzo fault. We also specify that we have no additional constraints to the timing for the activity of the Mondonico ridge.

Line285: What do you mean by transpressive fault architecture? Describe and reference it.

REPLY: we will rephrase this sentence. We will write that the inversion of the fault-slip data indicates that this fault was active during a transpressive phase. This is consistent with our observations about the Mondonico push-up and its kinematic relationship with the Martica-Boarezzo fault.

ACTUAL CHANGES TO THE TEXT: we deleted this sentence.

Line248: Describing not showcasing

REPLY: We will change it accordingly.

ACTUAL CHANGES TO THE TEXT: changed as suggested.

Line253: Are you sure the apparent left-lateral separation is due to normal fault displacement? It could also be the result of an irregular unconformity surface? Or a result of the southward continuation of the Mondonico push-up? Your restorations are dependent on how you have interpreted fault motion in the different blocks.

REPLY: Although some of the apparent left-lateral separation is related to Alpine deformation, we think that our interpretation is solid because of:

- A ca. 10 m wide fault zone has been observed in the field, including a well-developed fault gouge and kinematic indicators consistent with our interpretation;
- After 3D unfolding and restoration there is a residual offset along the Valganna Fault – the apparent 3 km left-lateral offset along the Valganna Fault cannot be traced further north – note that the base of the Volcanic Units are perfectly matching north of the MBF, with the only exception of the Mondonico Ridge.
- The Valganna Fault abruptly delimits, to the west, the extension of the intra-platform anoxic Middle Triassic facies (i.e., the Besano shales and Meride Fm.).

- The fault slip data collected along the Valganna Fault, after restoration nearest bedding orientation, clearly shows a normal activity.

In our view, the left lateral separation along the Valganna Fault cannot be related to an irregular unconformity. The middle Permian unconformity at the base of Anisian-Ladinian succession is an erosive surface with very little relief. The almost flat morphology of this erosive surface is inconsistent with a variation in thickness of about 200 meters, which we attribute to the activity of the Valganna Normal Fault.

We also exclude that the apparent left lateral separation is the result of the southern prosecution of the Mondonico Ridge because this structure is truncated by the middle Permian unconformity and no correlative offset can be observed north of the MBF.

ACTUAL CHANGES TO THE TEXT: These observations are outlined in section 4.1.1 and 4.1.2.

Line273: Describe the cross-cutting relationships
REPLY: Yes, we will add the description to the text.
ACTUAL CHANGES TO THE TEXT: the crosscut relationship between the Valganna Fault and the Matica – Boarezzo fault has been added in Section 4.1.2.

Line276: Structural stations not shown on Figure 3
REPLY: Yes, we will add those to the figures.
ACTUAL CHANGES TO THE TEXT: structural stations are shown in Figure 3

Line283-84: Figure order jumps from 5 to 7.
REPLY: We will change it accordingly.
ACTUAL CHANGES TO THE TEXT: We have fixed this.

Line 294: What do you mean by "entirely developed in the Anisian-Ladinian succession"? Do you have exposure of the top and bottom tip of the fault? A fault of this size would most likely originate at seismogenic depths. I think you mean it is exposed in the Anisian-Ladinian succession?
REPLY: Here we refer to the outcropping Valganna Fault in the location where the field photos were shot.
The Valganna Fault is a deeply rooted normal fault. Its upper tip is located withing the uppermost part of the Anisian-Ladinian succession: indeed the top of the Ladinian truncates the Valganna Fault. The lowermost outcropping segment of the Valganna Fault cuts through the basement and the Ganna Granitic Stock. A lower tip has not been observed.
ACTUAL CHANGES TO THE TEXT: we have changed the sentence to avoid misunderstandings.

Line331: "...and ZHe ages." Needs a reference here
REPLY AND ACTUAL CHANGES TO THE TEXT: We added appropriate references as, for instance, Reiners and Brandon, 2006.

Line335: Why isn't the alpha ejection accounted for and what are the implications for you ages? Need to discuss this further as VA07 has a problematic ages relative to VA05 and VA06.
REPLY: We account for the alpha-ejection as explained in the methods section (lines 189-190). However, as stated here, we do not account for the distribution of U and Th. The alpha-ejection correction assumes a homogeneous distribution of U and Th within zircons. Accounting for non-homogeneous distributions would require additional, non-routine measurements. Currently, U and Th are measured from bulk zircon grains after dissolution, aligning with the fact that He is also extracted from the entire grain. Measuring the spatial distribution of U and Th would necessitate laser-ablation ICP-MS profiling of elemental concentrations in the zircon. We will improve this sentence for clarity.
ACTUAL CHANGES TO THE TEXT: the revised text is: "Additionally, non-homogeneous U and Th distribution in zircon can result in an inaccurate alpha-ejection correction and in age dispersion (Hourigan et al., 2005). All these effects can occur at the same time making it difficult to separate them. In our samples, we did not correct for non-homogeneous U and Th distributions, as this would require additional, non-routine measurements."

Line350: Show where the previous age sample came from on Figure 3. This provides the reader with all the relevant information.
REPLY: On the map of the Figure 3, we will add the location of the cooling ages from previous studies mentioned in this line.
ACTUAL CHANGES TO THE TEXT: we updated Figure 3.

Line355: "volcanoclastic sample" – be specifis and consistent i.e. Early Permian effusive unit
REPLY: We will change it accordingly.
ACTUAL CHANGES TO THE TEXT: all the occurrences have been checked.

Line365: Ganna Granitic Stock needs an age range and a reference
REPLY: The age of the Ganna Granitic stock emplacement has been determined by Shalteger and Brack (2006) as stated in lines 84-85 "The emplacement of the Ganna Granitic complex occurred at 281.34 ± 0.48 Ma (Schaltegger and Brack, 2007)". We can add this reference and age here too.
ACTUAL CHANGES TO THE TEXT: added at line 372.

Line368: remove word far
REPLY: We will change it accordingly.
ACTUAL CHANGES TO THE TEXT: done

Line380: 2-3 km seems a very shallow intrusion depth and do not feel that your estimate should be based on your structural model. The closure temperature would suggest emplacement at depths of less than c. 5-6 km
REPLY: We will correct the text, and we will refer to the depth of emplacement of the stock as  estimated by Bakos et al. 1990.
ACTUAL CHANGES TO THE TEXT: see line 386.

Line402: Spelling structural
REPLY: We will correct it.
ACTUAL CHANGES TO THE TEXT: fixed

Line404: Refer to previous comments regarding the similarity in ages of VA08 and VA06. The thermal diffusion modelling does not rule out intrusion in beneath the northern block only that it would need to be 2-4 km below the sample locations.
REPLY: Yes, we agree. We will modify the text accordingly.
ACTUAL CHANGES TO THE TEXT: see line 396.

Line489: At present I would disagree with  "emplacement of a fault-bounded intrusive stock" – at present I do not feel you have adequately described how the and faulting are related in time.
REPLY: We partly agree. Indeed, we do not have a smoking gun to prove the absence of the stock north of the Marzio Fault but many clues point to it. We will explicit the degree of reliability of our conclusions in the text.
ACTUAL CHANGES TO THE TEXT: see line 396

Figure 1: Unit labels are confusing "Un." Normally abbreviates unconformity and not units as here. For consistency, Triassic unit colours should match those used in cross-sections.
REPLY: We will change it accordingly.
ACTUAL CHANGES TO THE TEXT: Figure 1 has been changed accordingly.

Figure 2: Study are should be shown
REPLY: We will change it accordingly.
ACTUAL CHANGES TO THE FIGURE: we added the study area in Figure 2

Figure 3: Consistency in unit naming – all units given age names except Effusive units. See text comments relating to consistency between unit naming in text and figure. Where are the structural stations? Show previous age determination location mentioned in the text.
REPLY: We will check for consistency and make necessary amendments. We will add the stations to the map.
ACTUAL CHANGES TO THE TEXT: the location of the structural stations is now shown; the effusive units does not have a traditional and well-established formational name. We prefer to indicate their chronostratigraphic position, instead.

Figure 4: Put horizontal scale on sections to show they are true scale. Caption change to "cross-sections in Figure 3". Where is Figure 4c on Figure 3? It is not obvious where this is as the fault trends do not seem to match Figure 3 very well?
REPLY: We will add a reference frame for locating figure 4c. Consider that the thickness map is created on the restored and unfolded geological model, thus changes in strike of the faults are expected.
ACTUAL CHANGES TO THE TEXT: We have changed the figure accordingly.

Figure 7: Strikes of dikes put on early-middle Permian for reference?
REPLY: Yes, we will add those.
ACTUAL CHANGES TO THE TEXT: we think that adding the strike of the dikes to Figure 7a could be misleading. The dikes are mostly subvertical with a NE-SW strike (see line 81). Those dikes are related to the emplacement of the Ganna Granitic Stock or to the early Permian tectonic phase – not to the successive early-middle Permian strike-slip tecontics. Thus, in the end, we decided not to add those to Figure 7.

REFEREE COMMENT 1

This manuscript analyzes the polyphase nature of the Permian-Triassic tectonics in a small region of the western Southern Alps. This issue has been the subject of numerous previous studies.

However, references to previous work of the last 15-20 years are almost completely missing from sections 1 and 2.

REPLY: To the best of our knowledge, we have thoroughly considered all relevant literature. We have made every effort to incorporate previous work where applicable. On July 12th, we requested the reviewer to provide examples of any missing references, but we have not received a response. As a result, we were unable to include or consider any additional works.

ACTUAL CHANGES TO THE TEXT: We were not able to include additional references.

COMMENT: The authors presents some new thermochronological data, which however do not provide essential elements for the resolution of the geological problem declared in the Introduction. As a result, the reader may feel that the conclusions presented are not adequately supported by the data shown. In my opinion, the rationale of the article is not adequately clear.

REPLY: We will carefully revise the text in light of this critique. While we agree that the thermochronologic data alone may not provide definitive constraints on the geological problem we address, they do offer valuable clues. When combined with geological and structural data, these insights contribute to a better understanding of the Variscan-Alpine transition. Additionally, we would like to highlight that the scientific hypotheses under investigation are clearly outlined in lines 36-54, and the specific contributions of this work are summarized in lines 54-60.

ACTUAL CHANGES TO THE TEXT: we revised the Section 1 but we confirm that the investigated hypothesis is clearly stated at lines 39-57.

COMMENT: A substantially stratigraphic/thermochronological Introduction and Geological Background sections are followed by an almost purely tectonic analysis, leaving the reader disoriented.

REPLY: We respectfully disagree. Constraints come from i) geological mapping of unconformity-bounded stratigraphic sequences: ii) structural analysis including restoration and thickness analysis and iii) thermochronologic data.

ACTUAL CHANGES TO THE TEXT: no changes.

COMMENT: Also, the title does not reflect the contents of the paper (the term "transpressive tectonics" which begins the title is never used in the text!)

REPLY: The term transpressive/transpressional has 11 occurrences in the text. However, we agree on that the title might not adequately reflect the content.

ACTUAL CHANGES TO THE TEXT:  We changed the title to: "Evidence for Multi-Rifting in the Variscan-Alpine Cycle Transition: Insights from the European western Southern Alps"

COMMENT:…and the abstract does not provide a concise and complete summary, and it is largely disconnected from the rest of the manuscript.
REPLY: We will rewrite the abstract considering this critique.
ACTUAL CHANGES TO THE TEXT: after careful consideration of this critique, we have concluded that we believe that the abstract does list the main questions, that we address in the introduction, and the conclusions, that report at the end of this manuscript. Thus, we improved the style of the abstract while maintaining the content.

REFEREE COMMENT 2

COMMENT: Firstly, there are some inconsistences between map and sections that must be fixed. Geological boundaries, fold axial traces and faults of the map do not correlate with the sections. Geological cross-sections are, moreover, a bit too rough. I suggest to take care of the geological data as well as to improve the quality of design. I also suggest to improve field (and cartographic) evidences of the intersection relationships between faults and intrusion and about the presence or not of faults synchronous with the volcanic activity.
REPLY: We will fix all the errors within the sections and the incongruences between the map and the sections.
ACTUAL CHANGES TO THE TEXT:  We have updated Figure 3 and better explained the crosscut relationship in Sections 4.1.1 and 4.1.2.

COMMENT: Regarding the thermochronology, data are sounding but the interpretation is quite weak. I do not understand how you may exclude the presence of buried intrusion below the northern block. 1D modelling is not enough. This is true for all samples, the distance of which from an intrusion is far to be demonstrated in the 3D space. It is also strange that one of the sample closest to the intrusion shows the youngest age. I lost the explanation for this.
REPLY: The 1D thermal models only helps to quantify the thermal effects of a shallowly emplaced granitic intrusion. While we do not believe there are significant inconsistencies in our interpretation, we acknowledge the need for greater clarity in our discussion of the data and will work to address this. Our interpretation of the thermochronologic data is based on the following observations:

- The cooling ages of samples VA05 and VA06 closely match the emplacement age of the Ganna granitic stock. These samples are located very close to the granitic intrusion, in the hangingwall of the Marzio normal fault that was active at the time of the granite emplacement.

- Sample VA08 exhibits cooling ages overlapping with the emplacement age of the Ganna granitic stock. This sample is located in the footwall of the Marzio normal fault, far from the granite.

- The cooling ages of sample VA07 span a broad range, from 250 to 170 Ma, that encompasses the time of the Ganna granitic intrusion and extends to the Middle Jurassic. This sample is very close to the Ganna granitic stock but in a position structurally lower than both the other samples and the granite. This sample is also proximal to a large syn-rift normal fault immediately to the east of the study area.

The 1D thermal model indicates that the thermal effect of a shallowly emplaced granitic intrusion drops rapidly both in space and time. By combining the observations with the 1D thermal model, we conclude that:

- The cooling ages of VA05 and VA06 may reflect post-magmatic cooling.
- The cooling ages of samples VA07 record both post-magmatic and rift-related cooling.
- The normal activity of the Marzio fault during the Permian led to uplift and exhumation of its footwall, recorded by sample VA08.
- The typical cooling pattern across a normal fault—where younger cooling ages are found in the footwall—has been overprinted by the thermal impact of the granitic intrusion.

ACTUAL CHANGES TO THE TEXT: In the discussion, we added a new section with title "the Permian thermal record", where we discuss out thermochronologic data based on the points listed in our reply above.

COMMENT: Another point that needs to be addressed is related to the limited consideration of other Alpine or surrounding sectors that shared similar or complementary tectonic evolution during the Mesozoic. I suggest widening the perspective, better accounting information from other Alpine sectors or the well-preserved Variscan section of the Sardinian-Corsica Batholith. Even the info from the middle-lower crust of the Ivrea zone must be implemented. From these areas the Authors may consider significant information to improve the discussion about the best fitting geodynamic models.
REPLY: We will integrate the discussion with the proper literature referred to the Permo-Variscan evolution of the Sardinian-Corsica area.
ACTUAL CHANGES TO THE TEXT:  We have added the comparison, as requested. see lines 555 and 557.

---

## Author Response (AR2)

**Replies to Reviewer (in blue)**

**Associate Editor: Virginia Toy**

The reviewers have considered your revised manuscript, and as you can also read, are happy that you addressed almost all their concerns. However, I do think that you should still , as indicated by reviewer #1, add some more discussion of the connection between surface and mid-deep crustal structures of the Ivrea Verbano area based on other published geochronological data.

Re: We revised the discussion section "5.2 The Variscan-Alpine cycle transition" in the light of your recommendations. We added the following text: "Although no early Permian magmatic activity is present north of the Marzio Fault, mantle-derived basic intrusions in the adjoining Ivrea-Verbano Zone to the west cut across the lower crust, linking early Permian magmatic activity in the Southern Alps to crustal thinning, astenospheric upwelling and magmatic underplating (e.g., Shaltegger and Brack, 2007). Unlike the shallow crustal rocks of the Varese area, the Ivrea Verbano Zone's lower crustal rocks were exposed to the surface through stepwise exhumation and tilting from the Jurassic through the middle-late Miocene (e.g., Wolff et al., 2012) with minimal impact on the study area."

We did a few additional minor changes through out the text including deleting additional spaces between words, correcting misspellings, and in the first paragraph of section 5.2, deleting a couple of redundancies.

**Reviewer 1: Anonymous Reviewer**

Suggestions for revision or reasons for rejection
(visible to the public if the article is accepted and published)
The issues raised in the previous review have been almost completely addressed in the reply and in the revised text. However, the thermochronological dataset offers limited constraints to support the tectonic model. Only one sample provides data clearly distinct from the age of the intrusion. Anyway, the data is now well displayed and discussed.
I would also like to point out that in the discussion no mention was made of the deep tectonic structures of the Ivrea-Verbano area, which could instead provide very strong constraints.The presence of multiple geochronological constraints from both intrusives and shear zones offers the chance to link the evolution of shallow and deep crustal sections. Therefore, I suggest to improve the treatment of this connection between surface and mid-deep crustal structures.

Re: Thank you for your suggestion. The Varese area indeed exposes near-surface products of the magmatic activity that also affected the Ivrea-Verbano Zone. We have revised the discussion in Section 5.2, 'The Variscan-Alpine Cycle Transition,' to include remarks on both the connections and differences between the Varese area and the Ivrea-Verbano Zone.

**Reviewer 2: Chris Morley**

No suggestions